# Potential Impact Protection of Polymer Matrix Composite Panels Using Polyurea Coatings

**DOI:** 10.3390/polym17030385

**Published:** 2025-01-31

**Authors:** Jide Williams, Joseph Hoffman, Paul Predecki, Maciej Kumosa

**Affiliations:** 1Walter Scott, Jr. College of Engineering, Scott, Bioengineering Building, Colorado State University, 700 Meridian Ave., Fort Collins, CO 80523, USA; 2Center for Novel High Voltage/Temperature Materials and Structures, University of Denver, 2155 E Wesley Avenue, Denver, CO 80208, USA; joe.hoffman@du.edu (J.H.); paul.predecki@du.edu (P.P.); maciej.kumosa@du.edu (M.K.)

**Keywords:** ballistic impact, polymer matrix composite plates, polyurea, numerical simulations

## Abstract

The protective effect of polyurea (PU) coatings on polymer matrix composite (PMC) panels subjected to high-velocity ballistic impacts, particularly as a potential replacement material for large power transformer (LPT) tanks, has not been extensively reported in the literature. This study addresses the gap by presenting a numerical investigation into the ballistic performance of PMC panels with PU coatings. Due to the complex nature and high cost of experimental testing, this research relies on finite element modeling to predict the panels’ responses under impact. Glass fiber/epoxy and carbon fiber/epoxy composite panels were tested individually and in hybrid configurations while being subjected to simulated 400 m/s steel projectile impacts. This study first investigates the impact damage evolution in uncoated panels, analyzing the arrest depth as a function of the panel thickness. It then evaluates the effect of PU coatings on the ballistic response. The results demonstrate that PU coatings are three times more effective in protecting both glass and carbon fiber panels from penetration compared to simply increasing the panel thickness. Additionally, the utilization of PU coatings led to a reduction in cost, mass, and thickness while still preventing penetration of the projectile in the models.

## 1. Introduction

Polymer matrix composites (PMCs) are being used for structural and ballistic protection applications, essentially for their beneficial specific properties [1]. For example, in the construction, automobile, and aerospace industries, the demands for lighter, stronger, and graded materials are increasing. Due in part to increasing acts of vandalism, the usage of composite materials is being considered for applications in the power industry for mitigating impact damage as the demand for the reliability of grid components becomes more critical. The present numerical study was conducted as a continuation of work exploring the use of PMCs as structural materials for large power transformer (LPT) tanks [2]. LPTs are a key component of the electrical grid, and their reliability is critical for health, safety, and economic prosperity. However, in the last ten years, certain LPTs have been targets of vandalism. Aside from the rifle attack on the Metcalf substation in California on 16 April 2013 [3], which caused roughly USD 16 million in damage, an analysis done by the *Wall Street Journal* reported that there were 274 cases of power grid vandalism in the United States between 2012 and 2014 [4,5]. J. Williams et al. [2] have proposed that PMCs can reduce the impact damage and other challenges resulting from vandalism.

The tank of a traditional LPT is typically made of 10 mm mild steel to ensure structural strength, cost-effectiveness, and durability [2]. Senthil et al. [6] evaluated the ballistic resistance of mild steel plates with thicknesses of 4.7, 6, 10, 12, 16, 20, and 25 mm, using a 7.62 mm diameter armor-piercing projectile at impact velocities between 800 and 850 m/s, and projectile obliquities ranging from 0° to 62°. Their study showed that the 10 mm mild steel was perforated across all tested speeds and obliquities. While these findings highlight the vulnerability of mild steel LPT tanks to vandalism by high-powered rifles, they also underscore the need for exploring superior alternative materials for improved ballistic protection.

The use of PMCs for ballistic impact protection has been widely studied both experimentally and numerically. J.J. Andrew et al. [7] provided a comprehensive review of various PMCs and the factors that influence their reaction to impacts, including material, geometry, environment, projectile type, and velocity. D. Long [8], using continuum-damage mechanics in a numerical simulation, modeled the effects of projectile size and velocity for impacts on a carbon fiber/epoxy panel. S. Chocron et al. [9] developed a numerical model and conducted experimental impact tests on carbon fiber-reinforced polymer (CFRP) panels for non-woven and woven architectures. N. Domun et al. [10] conducted ballistic impact tests on glass fiber-reinforced polymer (GFRP) composites and found that the ballistic performance could be improved by adding nanoparticles to the epoxy matrix with an attendant increase in cost.

Unidirectional (UD) PMCs provide a significant advantage for structural applications due to their superior strength in the fiber direction but are limited because of their low strain to failure [11]. Admittedly, loads that are applied transversely to the fibers in structures made of UD PMCs may cause catastrophic damage; nevertheless, UD laminates were modeled as a starting point in this study. In situations where impact damage from hailstones, vandalism, or loads applied normally to the impact surface are present, UD PMCs may not be adequate for use alone as a load-bearing structure.

Hybrid PMC structures have also been studied. In the experimental work done by P. Beaumont et al. [1] and B. Lusk et al. [12], the authors proposed a PMC laminate sandwich architecture consisting of two different composite types that offered the best impact resistance over several other composite architectures with an improvement of over 40 times. The laminate consisted of face sheets of 5 UD CFRP plies at the top and bottom of the laminate, with a core made up of 10 plies of UD GFRP. The improved impact resistance of the sandwich structure was attributed to the process of delamination at the CFRP/GFRP interfaces, which was shown to absorb much of the energy. However, the investigation was limited to quasi-static loads.

PU’s ballistic resistance capability is known to be due to its so-called ‘high-strain rate sensitivity’. Under high strain rates (10^5^/s to 10^7^/s), certain types of PU respond with high shear strength, which can be higher than that of steel [13,14]. The shear strength of PU in a ballistic event greatly depends on the thickness of the substrate [14]. In a high pressure and strain rate situation, PU’s shear strength can exceed that of most engineering materials used in protection technology. When used as a coating on metals and composites, an impact-resistant polymer should have a significant glass transition temperature well below operating temperatures of 20 to about 300 °C for LPTs. It has been shown by R. B. Bogoslovov et al. [15] and discussed by Y. Sun et al. [13] that for PU, the strain-rate-induced glass transition (the broad α-transition) leads to enhanced energy dissipation upon impact. This occurs when the deformation rate is comparable to the rate of motion of the soft segment in the PU chain, which, for several PUs, is of the order of 10^5^ to 10^7^/s at room temperature. At quasi-static strain rates (10^−2^/s), this α-transition occurs around −50 to −60 °C, well below LPT operating temperatures. For application in LPT tanks, PU could offer significant additional advantages in terms of durability and resilience for protection against environmental elements. It has an estimated 75-year outdoor life span [16], with good resistance to ultraviolet (UV) light.

PU has been used as a coating for several substrates, such as steel, aluminum, ceramics, etc. [14,16,17,18,19,20], but has only been recently studied as a coating for brittle composites such as CFRP to improve blast resistance [21]. An interesting observation made by Q. Liu et al. [21] was that placing the PU on the rear side of the CFRP textile structure provided significantly better spalling protection than having the coating on the front side. Zhang et al. [22] provided a comprehensive review of PU and its use in blast and impact protection for various substrates. For ballistic protection, the authors indicated that PU is generally placed on the front of the substrates. This approach in LPT tanks would avoid PU contact with the transformer fluid. The authors also identified certain disadvantages of protective PU coatings, such as difficulty in repairing damaged coatings, as well as spraying and coating problems due to uneven mixing, which can greatly influence coating effectiveness, in addition to the elevated costs of preparation and application. In a related study on the impact behavior of porcelain bushings on LPTs, it was shown that the bushings can be protected against impact when coated with polyurethane urea (LINEX) [23,24,25]. With the advent of big data and machine learning, convolutional neural networks have been used to predict the dynamic, cohesive properties of a continuously nanostructured copolymer such as PU [26].

In the present study, finite element (FE) analysis was employed to numerically assess the ballistic impact resistance of polymer matrix composite (PMC) materials, with a particular focus on the synergistic effects of polyurea (PU) coatings. Using versatile FE models built with Python scripts for the Abaqus Explicit solver, this study explored the ballistic performance of CFRP, GFRP, and hybrid composite panels with and without PU coatings. While experimental investigations have previously examined the ballistic protection offered by PU coatings on PMCs, this is, to the best of our knowledge, the first comprehensive numerical simulation that investigates this effect in detail.

In addition, our work addresses a critical gap in understanding the use of PMCs with PU coatings for ballistic impact protection, particularly in applications such as LPT tanks. Although previous studies have explored the ballistic and blast resistance of PMCs and PU coatings separately, their combined performance under high-velocity impacts has been insufficiently studied. The current research leverages a validated FE approach to provide a flexible and efficient method that reduces reliance on costly and time-consuming experimental testing. This novel numerical framework enables the exploration of the synergistic effects of PU coatings on PMC panels, offering insights into potential cost-effective and lightweight protective structures. The findings from this study not only contribute to the optimization of PMC panels for various structural applications but also provide a new pathway for designing composite materials for ballistic protection in fields where performance and cost are critical factors.

## 2. Materials and Failure Models

This study uses PMCs, specifically CFRP and GFRP laminates, as the primary materials for the simulations. The CFRP lamina properties were sourced from Simulia [27], and while these properties were not further verified in this study, they serve as a valid starting point for the analysis. In contrast, the GFRP properties were computed using Helius Composite software and its knowledge network [28]. These properties were developed from basic fiber and matrix properties such as the elastic moduli ($E_{f}=68\;GPa,\;E_{m}=3.5\;GPa)$, Poisson’s ratios ($\nu_{f}=0.25,\;\nu_{m}=0.35$), and the fiber volume fraction $V_{f}=0.7,$ where subscripts *f* and m refer to fiber and matrix, respectively. The values were obtained using the rule of mixtures.

Hybridization was explored using a combination of CFRP and GFRP laminas. Cross-ply CFRP and GFRP laminates were analyzed individually and as a hybrid composite. The material properties of GFRP and CFRP UD laminas are summarized in Table 1. The lamina properties for the CFRP provided by Simulia [27] did not specify the exact carbon fiber type they used to model the H. Kasano experiment [29]. A representation of the finite element model used is shown in Figure 1 and is discussed in later sections. To evaluate the ballistic impact resistance of these materials, a PU coating was also applied to the composite panels in simulations. The PU properties were modeled using a virtual user-defined field (VUSDFLD) for impact analysis, and a Prony series was used to describe the elastic–plastic response of the PU under ballistic loading.

The modeling procedure used is shown in Figure 2. Here, it is important to note that the model development section includes all the failure models used to describe the onset and progress of material failure that are described later in Section 2. Meanwhile, the composite lamina failure and the PU failure are all controlled by subroutines, which are integrated into the Abaqus Explicit solver. When the Python code containing all the commands is executed, all the subroutines within the Python code are called and are executed simultaneously.

The user subroutine for composite progressive damage was made available by Simulia [27] in their documentation. It was enhanced by utilizing Hashin’s damage model for composites [30] for the initiation and evolution of the through-thickness damage to the fiber and matrix for PMCs. In the subroutine, the Hashin damage criterion was specified for the fiber failure mode, while the Puck criterion [27,31] was applied to the matrix damage modes. The relationship between fiber and matrix damage utilized in the user subroutine was given by Simulia [27] as follows:(1)$$Fiber\;damage\left(tension\right):d={\left(\frac{\sigma_{11}}{X_{1tf}}\right)}^{2}+{\left(\frac{\tau_{12}}{S_{12f}}\right)}^{2}+{\left(\frac{\tau_{13}}{S_{13f}}\right)}^{2}$$(2)$$Fiber\;damage\left(compression\right):d=\frac{\left|\sigma_{11}\right|}{X_{1cf}}$$

Matrix damage in tension and compression (Puck)(3)$$d=\left[{\left(\frac{\sigma_{11}}{{2X}_{1tm}}\right)}^{2}+\frac{\sigma_{22}^{2}}{X_{2tm}.X_{2cm}}+{\left(\frac{\tau_{12}}{S_{12m}}\right)}^{2}\right]+\sigma_{22}\left(\frac{1}{X_{2tm}}+\frac{1}{X_{2cm}}\right)$$
where $\sigma_{ij}\;and\;\tau_{ij}$ are the effective normal and shear stress tensors and $X_{1tf},\;{{\;X}_{1cf},\;\;S}_{12f},\;and\;S_{13f}$ are the fiber tensile, compressive, and shear strengths. $X_{1tm}\;and\;X_{2tm}$ are the matrix tensile failure stresses in the 1 and 2 directions, $X_{2cm}\;$ is the matrix compressive failure stress in the 2 direction, and $S_{12m}$ is the matrix failure shear stress.

The justification for using UD PMCs for this study is based on the primary type of loading expected when UD PMCs are used for LPT tanks. For such a tank having the shape of a rectangular prism made up of UD PMC panels joined at the edges, the primary loading (stress) these panels experience when in service was shown to be in-plane tensile [32]. Moreover, the Hashin and Puck failure criteria are well known for describing the failure behavior of UD PMCs [30].

For PU, a single composition was chosen to demonstrate the effectiveness of PU coatings in attenuating impact on the PMC panels. The PU used in this work has a composition ratio of 4 parts by weight of Versalink P-1000 diamine to 1 part of Isonate 143L isocyanate and was selected for two reasons. Firstly, this composition has been extensively studied by research groups funded by the US Office of Naval Research at UCLA and UCSD, with the 4:1 ratio being generally considered optimal for impact applications [33]. Secondly, a Prony series for the viscoelastic response of this PU composition was available [34]. In addition, this composition ratio is close to being stoichiometric since the equivalent weights are 575–625 g/equivalent for the Versalink P-1000 [35] and 144.5 g/equivalent for the Isonate 143L [36], resulting in a mass ratio of 3.98–4.35:1 for a stoichiometric ratio of 1:1 for the amine to isocyanate groups. Thus, the 4:1 mass ratio falls within this stoichiometric mass ratio range.

The material properties of the particular PU investigated by C.A. Gamez [34] are listed in Table 2. The time-dependent behavior of PU is attributed to the material’s hard and soft segments, i.e., the motion of the chain at different time scales. PU has been modeled as a linear viscoelastic isotropic solid material by C.A. Gamez [34] using the material’s shear relaxation modulus *G*. A Prony series was used in [21] and in this work to model the relaxation of a polymer consisting of *n* decaying exponentials in a generalized Maxwell model. Equations (4) and (5) describe the time-dependent shear and bulk moduli of the PU under high strain rates [37].(4)$$G_{t}=\left[G_{\infty }+\sum_{k=1}^{n}g_{k}G_{0}.exp\;(-\frac{t}{\tau_{k}})\right]\;\mathrm{where}\;g_{k}=\frac{G_{k}}{G_{0}}$$(5)$$K_{t}=\left[K_{\infty }+\sum_{k=1}^{n}k_{k}K_{0}.exp\;(-\frac{t}{\tau_{k}})\right]\;\mathrm{where}\;k_{k}=\frac{K_{k}}{K_{0}}$$

$G_{t}\;and\;K_{t}\;are\;the\;shear\;and\;bulk\;moduli\;as\;a\;function\;of\;time,\;g_{k}\;$ is the ratio of the shear moduli for the *k*-*th* relaxation component, defined as $g_{k}=\frac{G_{k}}{G_{0}}.$ $G_{k}\;$ is the shear modulus contribution from the *k*-*th* term in the Prony series. $G_{0}$ is the initial shear modulus (at time *t* = 0). Likewise, $k_{k}\;$ is the ratio of the bulk moduli for the *k*-*th* relaxation component, defined as $k_{k}=\frac{K_{k}}{k_{0}}$, where $K_{k}\;$ is the bulk modulus contribution from the *k*-*th* term in the Prony series. The term $\exp\left(-\frac{t}{\tau_{k}}\right)\;$ is the exponential decay term showing time dependence and $\tau_{k}$ is the relaxation time of the kth element, which is the time it takes to ‘relax’ the stress to about 37% (1/e where *e* is Euler’s number) of the initial applied stress. $G_{0}$ and $K_{0}$ are the shear and bulk moduli at time t = 0, and $G_{\infty }\;and\;K_{\infty }$ are the shear and bulk moduli of the first element in the generalized Maxwell model. In this work, 14 terms of the Prony series, provided by the experimental work of [34], were used in the simulation.

Finally, a virtual user-defined field (VUSDFLD) [38] subroutine was used to assess the response of the PU. VUSDFLD is a custom constitutive model used for modeling the high strain rate behavior of the PU. This subroutine computes the strain in the material at each time step using the Prony series and then compares it with a critical strain value at which the material fails. With limited published data on the critical strain values for PU at the strain rates under study, we selected a failure value of 0.25 to perform element elimination from the model. We believe this value of 0.25 is conservative, given the stress–strain curves for various PUs that we reviewed [14,39,40].

For the 4340 steel projectile, the material model and the calibrated Johnson–Cook (J-C) parameters, shown in Table 3, were used. The material response for this steel is linear elastic up to its yield strength, after which it deforms plastically until failure [38]. The yield stress (${\bar{\sigma }}_{Y}$) and the strain at failure (${\bar{\varepsilon }}_{f})$ [41] are expressed by Equations (6) and (7). The Johnson–Cook model is proven and validated widely both numerically and experimentally, and it is the standard in the modeling response of metallic materials under high strain rate conditions [42].(6)$${\bar{\sigma }}_{Y}\left(A,B,n,C,\dot{\bar{\varepsilon_{0}},}T_{m},T_{0},m\right)=\left[A+B{\left({\bar{\varepsilon }}_{pl}\right)}^{n}\right]\left[1+C\;ln\frac{{\dot{\bar{\varepsilon }}}_{pl}}{{\dot{\bar{\varepsilon }}}_{0}}\right]\left[1-{\left(\frac{T-T_{0}}{T_{m}-T_{0}}\right)}^{m}\right]$$(7)$${\bar{\varepsilon }}_{f}\left(d_{1},d_{2},d_{3},d_{4},d_{5},\dot{\bar{\varepsilon_{0}},}T_{m},T_{0}\right)=\left[d_{1}+d_{2}\;exp\;(d_{3}\eta )\right]\left[1+d_{4}\ln\left(\frac{{\dot{\bar{\varepsilon }}}_{pl}}{{\dot{\bar{\varepsilon }}}_{0}}\right)\right]\left(1+d_{5}\frac{T-T_{0}}{T_{m}-T_{0}}\right)$$

In Equations (6) and (7), *η* = −*p*⁄*q* represents the triaxiality, defined as the hydrostatic pressure p divided by the deviatoric stress (von Mises stress) q; $\dot{\bar{\varepsilon_{0}}}$ is the reference strain rate; *T* is the attained temperature of the material; *T_m_* is the melting point; *T*_0_ is the ambient temperature or the reference temperature; ${\bar{\varepsilon }}_{pl}$ is the plastic strain and ${\dot{\bar{\varepsilon \;}}}_{pl}$ the plastic strain rate; *A* is a constant, which is the yield strength of the material at reference conditions; *B* is a constant, which controls the rate of increase in yield stress of the metal with increasing plastic strain; and *C* is the viscous effect constant, which controls the strain rate sensitivity of the material. *n* is the strain hardening exponent, *m* is the thermal softening exponent, and $d_{1},d_{2},\;d_{3},\;d_{4},\;{and\;d}_{5\;}$ are all damage parameters. Data for this steel, which constitutes most of the values in Table 3, were obtained from [43]. The damage regime of the projectile was not considered because the projectile can be considered as a “partially rigid material” when compared to the PMC plies; therefore, $d_{1},d_{2},\;d_{3},\;d_{4},\;{and\;d}_{5\;}$ were ignored in the simulation, making only Equation (6) relevant to the study. Equation (7) is retained here for cases in which the target is sufficiently strong to produce damage in the projectile.

**Table 3 polymers-17-00385-t003:** Johnson–Cook properties for 4340 steel projectiles.

Description	Notation	Property [44]
Modulus of elasticity	E (GPa)	208
Poisson’s ratio	ν	0.3
Density	ρ kgm3	7830
Yield Stress	$A\;\left(MPa\right)$	792
Strain hardening constant B and exponent n	$B\;\left(MPa\right)$ n	510 0.26
Viscous effect constant	C	0.014
Thermal softening constant	m	1.03
Reference strain rate	$\;\dot{\bar{\varepsilon_{0}}}$	0.000333
Melting temperature	$T_{m\;\;}\left(K\right)$	1700
Transition temperature	$T_{0}\;\left(K\right)$	571

During impact, the high plastic strain rate causes the dissipation of energy as heat, which raises the temperature (T) of each of the impacted elements in the model. The temperature rise is exclusive to each element and there is no heat conduction assumed between elements. The inelastic heat fraction applied in the simulation was 0.9; that is, 90% of the energy dissipated by the plastic deformation was assumed to be converted to heat.

## 3. Independent Verification of FE Models

To confirm the validity of the PMC model, a comparison was made to an experiment conducted by H. Kasano [29]. In that study, a 5 mm diameter spherical steel projectile of a mass of 0.51 g was fired at three different velocities at a multi-ply composite panel, and the residual velocities were measured. The composite panel was a [0, 90, 0] _ns_ laminate with 12 or 18 layers of CFRP, in which *n* (the number of repetitions or stacks) = 2 or 3, respectively, and the panel thickness ranged from 1.8 mm to 2.7 mm. For this validation, the 12-layer (n = 2) experiment was used. A 200 × 200 mm CFRP panel was simulated using a target area of 50 × 50 mm. Taking advantage of symmetry, a quarter model was developed to reduce the computation costs, as seen in Figure 1a,b. Note that while the color schemes are different, the geometry is the same, and Figure 1b is a zoomed-in view of Figure 1a with a different perspective. The architecture, plies, and thicknesses of the model followed those described in H. Kasano’s experimental study [29]. The target region, also called the region of interest, was divided into sub-laminates, where each [0, 90, 0] sub-laminate was separated by a cohesive zone (interply). The cohesive zones were modeled with cohesive elements (COH3D8), where a 3D 8-node element models the behavior of interfaces or bonds between the plies subjected to normal and tangential stresses. The maximum degradation energy, which is the energy required to fully degrade the interface, was set to 1 kJ.

The region of the plate outside the target area, the shell region in Figure 1a,b was modeled with shell elements (S4R) to reduce the computational cost. The mesh size of this region is insignificant and is left to the solver to automatically generate. The target region was represented with first-order C3D8R 3D continuum elements [27] (layer-by-layer) to capture the damage in detail. The element’s mesh size was initially set at 1 mm × 1 mm × 0.15 mm thick. The edges of the plate were fixed in place. To make the model more versatile, a Python script was used, in which the geometry parameters, mesh parameters, contact parameters, speed of the projectile, and boundary conditions were all defined as variables.

The exact properties of the steel projectile used in the experiment in [29] were not stated. In our work, cold-rolled 4340 steel was used to model the projectile as a partially deformable spherical solid [27]. The projectile yield followed the Johnson–Cook parameters listed in Table 3. These complete parameters may be used to simulate both the initiation and propagation of the damage in the projectile under elastoplastic conditions. However, this study limits the projectile to a partially elastic material in order to simplify the model, as was discussed in Section 2.

A mesh sensitivity study was conducted for the target, with mesh sizes of between 0.3 and 1.0 mm, and for the projectile, with mesh sizes of between 0.35mm and 0.55mm. A good correlation was found with the experimental result at mesh sizes of 0.85 mm in the planar directions and at one element per ply in the thickness direction (0.15 mm) for the target and 0.4 mm for the projectile. The comparison of our simulation results with the experiment of H. Kasano [29] is shown in Figure 3.

The blue curve in Figure 3 is the fitted curve from [29] for the 12-layer panel utilizing the following function:(8)$$V_{R}=\alpha \sqrt{{V_{i}}^{2}-{V_{bl}}^{2}}$$
where $V_{i}\;$ is the impact velocity, $V_{bl}$ is the ballistic limit (the highest velocity at which total penetration is just prevented), and α is a function of the mass of the projectile and the masses of the fragments released from the plate upon impact (for more information, see [29]). Certain points along that curve are specified for comparison with the model where those same velocities were simulated. A curve using the same parameters as the experiment (i.e., with α = 0.9) is fitted to those points and displayed in the top curve (the orange curve) of Figure 3. While the variation in the residual velocities decreases as the impact velocity increases, the difference in the residual kinetic energies of the projectile between the experiment and the simulation stays roughly constant. Using either measure, the simulation produces a conservative estimate of the energy dissipated by the composite plates, and thus, the experimental results should be better than those predicted by the modeling.

## 4. Impact Response of CFRP and GFRP Laminates

Next, the impact resistance of the CFRP and GFRP composites was evaluated using the same model that was verified in the previous section. All the simulations utilized individual unidirectional 0.2 mm plies in a [0/90/0] architecture. Thus, each [0/90/0] sub-laminate was 0.6 mm thick. The target and plate lateral dimensions were the same as those used in the PMC verification. Again, C3D8R elements were used in the target area, and S4R elements were used in the non-target area. Lateral dimensions of 0.85 mm and thickness dimensions of 0.2 mm were used for the PMC target’s element mesh sizes. The thickness dimension of the PMC mesh corresponded to one element per ply. The edges of the plates were fixed as was done for the validation.

The details of the boundary conditions are summarized in Figure 4 below.

Initially, it was found that both CFRP and GFRP laminates having 18 sub-laminate layers of total thickness 10.8 mm were penetrated in the simulation by 5mm 4340 spherical steel projectiles traveling at 400 m/s. This total thickness is comparable to the standard thickness of LPT tanks [2]. The thicknesses for both the CFRP and the GFRP laminates were then increased by adding additional sub-laminates to find a thickness (termed the Structure Thickness in Table 4) that was slightly greater than that which would just prevent penetration. Subsequently, a somewhat larger thickness of 27.6 mm (termed the Constant Structure Thickness in Table 4) was utilized to allow for the comparison of penetration depths and to see whether penetration depths would significantly change if the thicknesses were increased beyond the minimum penetration prevention thickness. See Figure 5a,b for simulation schematics. The results for both thickness types are presented in Table 4a.

Following the proposal made by P. Beaumont et al. [1], two hybrid architectures of CFRP/GFRP and GFRP/CFRP (where the projectile impacts the first material listed before the second, and both components were of equal thickness) were evaluated. In both cases, the initial 10.8 mm thick hybrid plates were unable to prevent full penetration by the 400 m/s steel projectile. The same process to estimate the structure thicknesses and arrest depths, as discussed in the prior paragraph, was then utilized for these hybrid structures. The results are presented in Table 4b.

The results in Table 4 show that for CFRP, the arrest depth for the thinner 19.8 mm structure thickness is quite similar to that for the thicker 27.6 mm structure: 17.1 mm and 17.3 mm, respectively. Likewise, for GFRP, the arrest depths for the thinner 22.2 mm structure thickness are quite similar to that of the thicker 27.6 mm structure: 21.2 mm and 20.8 mm, respectively. It can be seen that when the thickness was increased by approximately 30–40%, the arrest depth did not change materially. This suggests that we can simulate the projectile arrest with a plate thickness relatively close to the arrest distance. However, this might create a bulge at the backface of the plate containing delaminations, which could affect the projectile arrest process. The projectile would be stopped, but the backface would be severely damaged, leading to a partial arrest. Hybrid CFRP + GFRP laminates had almost the same arrest depths, regardless of their order (Table 4b), and were of comparable magnitude to those of the non-hybrid laminates. Thus, the significant improvements from hybridization, as reported by [1,12], were not observed in our simulations.

## 5. The Effect of PU Coating

The modeling of the PMCs with PU utilized the same PMC mesh sizes and a mesh size of 0.3 mm for the PU in all directions. The same bond energy used for the simulation of the bonding between two PMC laminas was used for the simulation between PMC plies and the PU coating. With the addition of 3 mm of PU to the original 10.8 mm thick plates, the PU/CFRP plate successfully arrested the 400 m/s projectile, as shown in Table 5. Figure 6 illustrates the simulated results. However, with the GFRP laminate, the addition of the PU only allowed for the partial arrest of the projectile since the arrest depth of 13.77 mm was almost the same as the laminate depth of 13.8 mm. In this case, the coating thickness or the GFRP thickness would need to be increased slightly to achieve full arrest. A hybrid combination of PU/CFRP/GFRP with a total thickness of 13.8 mm also successfully arrested the projectile. A comparison of Table 5 with Table 4 reveals the significant effect of the 3 mm PU coating on CFRP and GFRP laminates.

Thus, by applying a 3 mm coating of PU, the thicknesses of these laminates to just prevent penetration were significantly reduced compared with those of the corresponding uncoated laminates.

These results show that PU could play a major role in the impact protection of advanced PMCs, for example, in the protection of LPT tanks. To further demonstrate this effect, Figure 7 and Figure 8 are presented. We analyzed the damage formation process in the PU/CFRP/GFRP laminate (Figure 7) from the onset of the projectile impact until the projectile was fully arrested inside the laminate. Eight stages of damage progression between 0 and 150 microseconds are shown. Figure 8 shows the velocities of the projectile in the hybrid laminate with and without the PU coating.

Several important observations can be made when analyzing the data presented in Figure 7 and Figure 8. First, the projectile decelerations (change in velocities) are markedly different between the coated and uncoated composites in Figure 8. The reduction in the velocity of the projectile in the coating is significantly larger than in the composite. For example, the time taken for the projectile to decelerate from 400 to 200 m/s with PU is approximately 15 microseconds, while the same deceleration without PU takes about 45 microseconds, which is three times longer. At somewhere between 10 and 20 microseconds, the projectile has just penetrated the PU coating (Figure 7). Thus, the majority of the 400 to 200 m/s deceleration has resulted from the beneficial effect of the PU coating. Note that while the coated laminate in Figure 8 fully arrested the projectile and brought its velocity to zero, the projectile retained an exit velocity of about 200 m/s after penetrating the uncoated hybrid laminate.

To determine the approximate costs of the laminates, estimated costs of USD 25, USD 5, and USD 22.50/kg and densities of 1750, 1870, and 1071 kg/m^3^ for CFRP, GFRP, and PU, respectively, were used along with the simulated results from Table 4 and Table 5. Table 6 summarizes the modeled thicknesses to prevent penetration (the structure thicknesses), the costs/m^2^, and the masses/m^2^ for all six material combinations studied (where/m^2^ is for the indicated thickness). It is clear from this table that adding a 3 mm coating of PU reduces the thickness, cost, and mass required to prevent penetration for any of the substrate materials.

## 6. Conclusions

This study aimed to numerically assess the ballistic performance of polymer matrix composite (PMC) panels with polyurea (PU) coatings, particularly for their potential application in large power transformer (LPT) tanks. The primary objectives were to (1) evaluate the impact resistance of CFRP and GFRP laminates, (2) investigate the effectiveness of hybrid composite configurations, and (3) explore the benefits of PU coatings in enhancing ballistic protection while reducing the cost, mass, and thickness. The results show that the PU coatings significantly improve the ballistic resistance of PMC panels, proving to be three times more efficient than when increasing the panel’s thickness alone. Hybrid laminates combining CFRP and GFRP with PU coatings demonstrated an improved performance, offering a balance of strength and cost-effectiveness. Moreover, the addition of PU coatings reduced both the mass and cost of the panels while providing comparable or superior protection compared with uncoated composites.

These findings offer valuable insights into the potential use of PMC panels with PU coatings for LPT tank protection, presenting a promising alternative to traditional, heavier materials. The numerical framework developed in this study provides a reliable tool for initial material selection and design optimization in the development of protective structures for the power grid. While future work can explore additional projectile velocities and include experimental validation to confirm the recommended design, the results presented here demonstrate that PU coatings enhance ballistic resistance while simultaneously reducing the mass, thickness, and cost. This marks a significant advancement in material selection and design for protective applications.

## Figures and Tables

**Figure 1 polymers-17-00385-f001:**
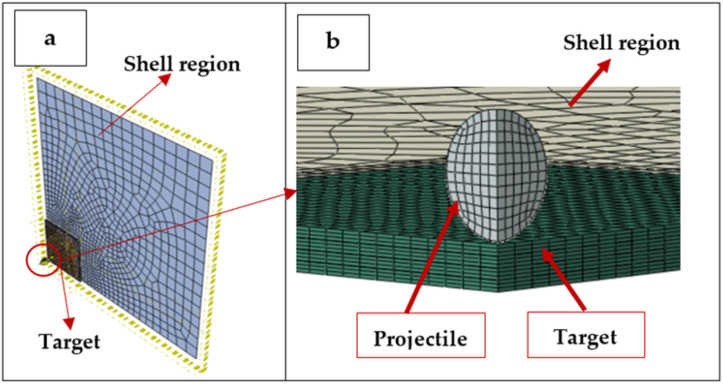
Numerical model of a composite plate under impact: (**a**) plate and target and (**b**) representation of target and projectile—not to scale.

**Figure 2 polymers-17-00385-f002:**
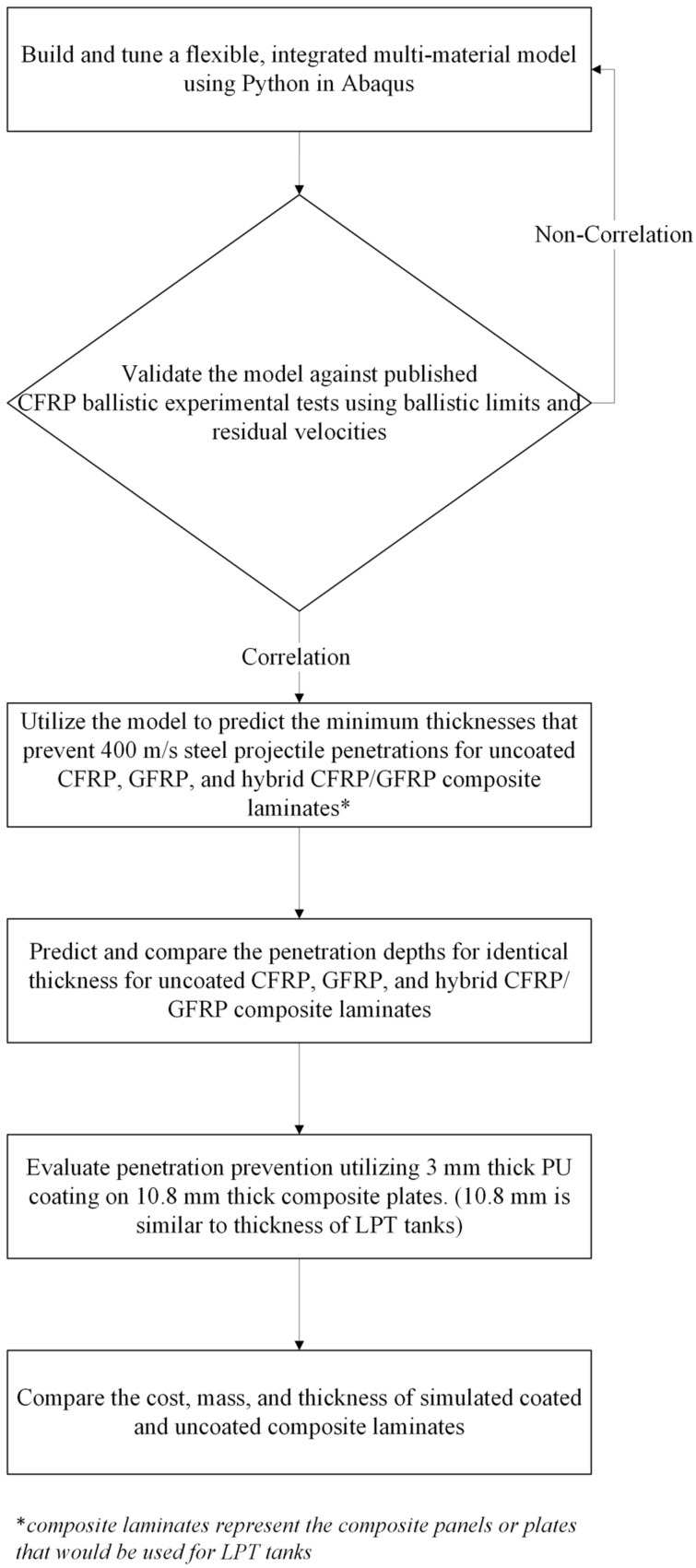
Flowchart describing the strategy for implementing the FEM and recording the penetration depths.

**Figure 3 polymers-17-00385-f003:**
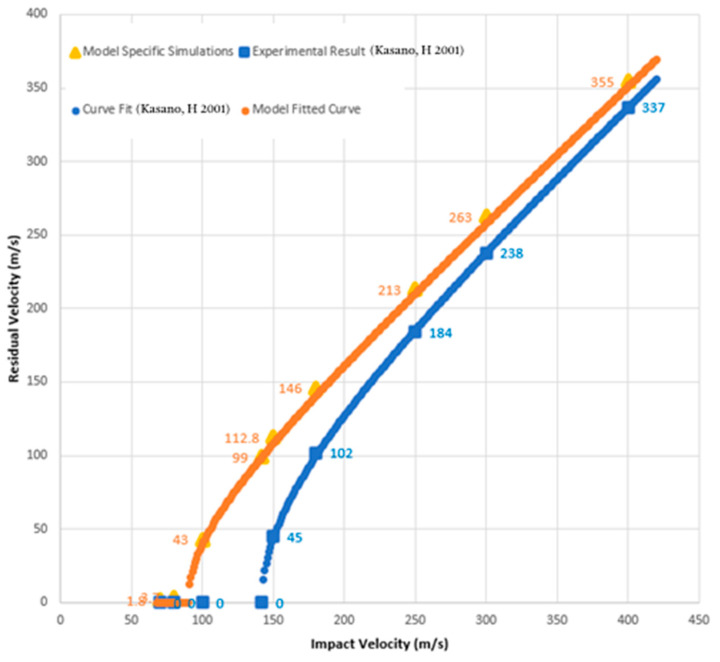
Comparison of the CFRP model with experimental results from Kasano, H 2001 [29].

**Figure 4 polymers-17-00385-f004:**
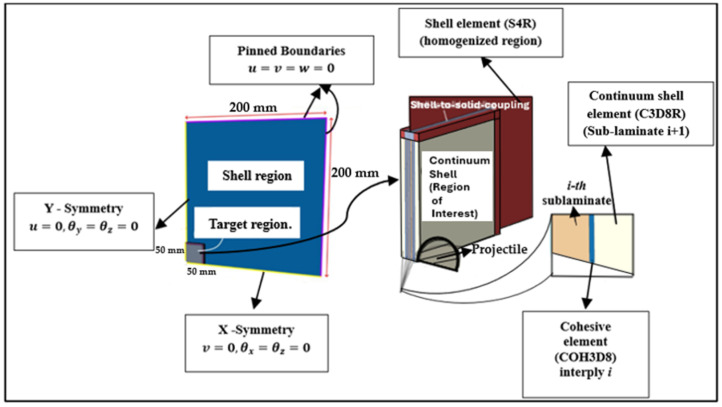
Schematic summary of model boundary conditions as related to model geometry.

**Figure 5 polymers-17-00385-f005:**
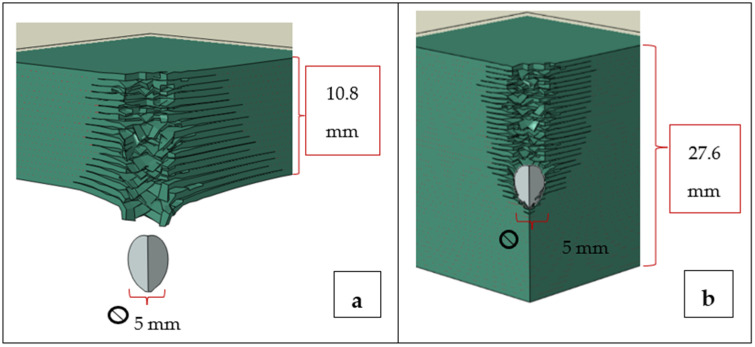
Penetrations of and arrests in CFRP plates using a 400 m/s projectile: (**a**) penetration in a 10.8 mm thick plate and (**b**) arrest in a 27.6 mm thick plate.

**Figure 6 polymers-17-00385-f006:**
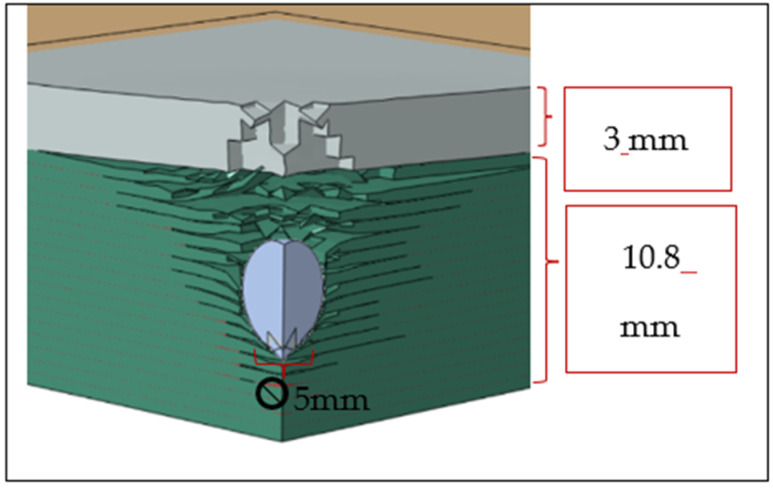
Penetration of and arrest in a 10.8 mm CFRP plate coated with 3 mm of PU using a 400 m/s 4340 steel projectile.

**Figure 7 polymers-17-00385-f007:**
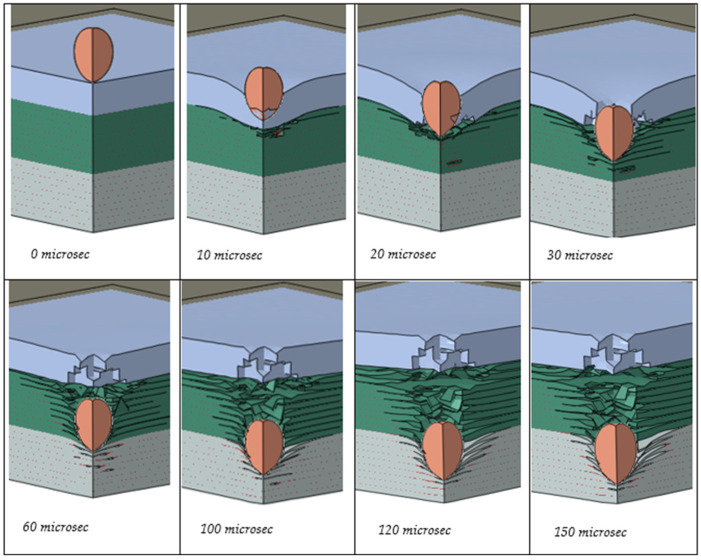
Progressive penetration of the projectile through the coated hybrid.

**Figure 8 polymers-17-00385-f008:**
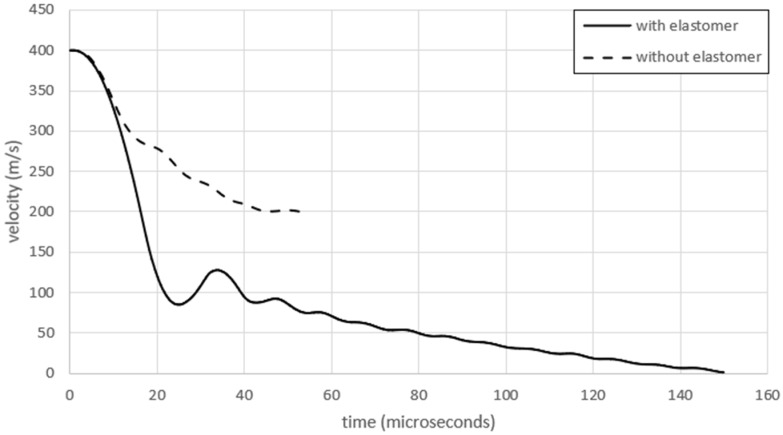
Velocities versus time after impact for the hybrid laminate with and without a 3 mm PU coating.

**Table 1 polymers-17-00385-t001:** Properties of unidirectional (UD) CFRP and GFRP laminas. The subscripts 1, 2, and 3 refer to the fiber direction, the transverse in-panel direction, and the normal to the lamina direction, respectively. Subscripts t and c refer to tension and compression, respectively.

Description		CFRP [27]	GFRP [28]
Young’s modulus, $E_{11}$	(GPa)	235	53
Young’s modulus, $E_{22}$	(GPa)	17	12.4
Young’s modulus, $E_{33}$	(GPa)	17	12.4
Poisson’s ratio $\nu_{12}$		0.32	0.26
Poisson’s ratio $\nu_{13}$		0.32	0.26
Poisson’s ratio $\nu_{23}$	(GPa)	0.45	0.45
Shear modulus, $G_{12}$	(GPa)	4.5	4.18
Shear modulus, $G_{13}$	(GPa)	4.5	4.18
Shear modulus, $G_{23}$	(GPa)	2.5	4.27
Tensile Failure Stress, $X_{1t}$	(MPa)	3900	2750
Comp. Failure Stress, $X_{1c}$	(MPa)	2400	1470
Tensile Failure Stress, $X_{2t}$	(MPa)	111	59.6
Comp. Failure Stress, $X_{2c}$	(MPa)	290	293
Tensile Failure Stress, $X_{3t}$	(MPa)	50	59.6
Comp. Failure Stress, $X_{3c}$	(MPa)	290	293
Failure Shear Stress, $S_{12}$	(MPa)	120	122.6
Failure Shear Stress, $S_{13}$	(MPa)	137	124
Failure Shear Stress, $S_{23}$	(MPa)	90	124

**Table 2 polymers-17-00385-t002:** Material properties for polyurea [34] at room temperature.

Density (kg/m^3^)			1071
Elastic Modulus (GPa)			1.084
Bulk Modulus (GPa)			4.54
Poisson’s Ratio			0.486
Viscous Properties (Prony Series) [34]			
$g_{k}$	$k_{k}$	$\tau_{k}$ (s)	
0.03691	0.03691	1.00 × 10 ^−13^	
0.03691	0.03691	5.00 × 10 ^−13^	
0.03691	0.03691	1.00 × 10 ^−12^	
4.10 × 10 ^−17^	4.10 × 10 ^−17^	1.00 × 10 ^−11^	
0.222841	0.222841	1.00 × 10 ^−10^	
0.176243	0.176243	1.00 × 10 ^−9^	
0.116726	0.116726	1.00 × 10 ^−8^	
0.092643	0.092643	1.00 × 10 ^−7^	
0.063106	0.063106	1.00 × 10 ^−6^	
0.042889	0.042889	1.00 × 10 ^−5^	
0.037371	0.037371	0.0001	
0.019091	0.019091	0.001	
0.016129	0.016129	0.01	
0.010039	0.010039	0.1	

**Table 4 polymers-17-00385-t004:** Penetration results from 400 m/s tests with 4340 steel projectiles: (**a**) CFRP and GFRP laminates and (**b**) CFRP and GFRP hybrid laminates. Arrest depth is the depth at which penetration velocity is reduced to zero.

**Target** **(Individual PMCs)** **a**	**Structure** **Thickness** **(mm)**	**Arrest** **Depth I** **(mm)**	**Constant** **Structure Thickness (mm)**	**Arrest Depth II** **(mm)**
CFRP	19.8	17.1	27.6	17.3
GFRP	22.2	21.2	27.6	20.8
**Target** **(Hybrid PMCs)** **b**	**Structure** **Thickness** **(mm)**	**Arrest** **Depth I** **(mm)**	**Constant** **Structure Thickness (mm)**	**Arrest Depth II** **(mm)**
CFRP + GFRP	19.8	17.6	27.6	17.5
GFRP + CFRP	21.6	18.7	27.6	18.8

**Table 5 polymers-17-00385-t005:** Penetration results for CFRP, GFRP, and hybrid laminates with PU coatings subjected to 400 m/s 4340 steel projectiles.

Target (Individual PMCs with Coating)	Structure Thickness (mm)	Arrest Depth Depth (mm)
3 mm PU + 10.8 mm CFRP	13.8	12.05
3 mm PU + 10.8 mm GFRP	13.8	13.77 (partial penetration)
3 mm PU + 5.4 mm CFRP + 5.4 mm GFRP	13.8	12

**Table 6 polymers-17-00385-t006:** Thickness, cost, and mass of material combinations required to prevent ballistic penetration: CFRP, GFRP, and hybrid composites with and without PU coating.

Material	Required Thickness (mm)	Cost/Square Meter (USD)	Mass (kg)/Square Meter (kg)
CFRP	19.8	866.25	34.7
GFRP	22.2	213.18	42.6
CFRP+GFRP	19.8	525.69	35.8
PU+CFRP	13.8	544.79	22.1
PU+GFRP	13.8 *	173.27	23.4
PU+CFRP+GFRP	13.8	359.03	22.8

* Partial penetration.

## Data Availability

Data may be provided by the corresponding author upon email request.

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
