# Peer review of "Potential Impact Protection of Polymer Matrix Composite Panels Using Polyurea Coatings"

_polymers, 2025, doi:10.3390/polym17030385_

Round 1

Reviewer 1 Report (Previous Reviewer 1)

Comments and Suggestions for Authors

I reviewed this manuscript before and I recommended acceptance as it is. 

Author Response

Thank you very much for your valuable review which helped to significantly improve the paper.

Reviewer 2 Report (New Reviewer)

Comments and Suggestions for Authors

The current work covers a somewhat tough topic to publish. The scope is broad enough but this is for the journal to decide. Below my main comments:

L 46: please use et al. format. See also e.g. L 100.

A general reader needs some more information on how chemically/physicall the coating is interacting with the surface (or typical surface). E.g. add a graphical representation.

More details need to be given of the fabrication of the material configuration, so how can other reproduce it?

Please explain in more detail the simulation approach? Add a flow chart.

Please make sure all abbreviations are explained.

Equation (6) and (7): how reliable are these?

Figure 3: 400 m/s; could this be a variable?

Author Response

The current work covers a somewhat tough topic to publish. The scope is broad enough, but this is for the journal to decide. Below my main comments:

L 46: please use et al. format. See also e.g. L 100.

Response: These has been appropriately adopted in the latest version of the manuscript.

A general reader needs some more information on how chemically/physicall the coating is interacting with the surface (or typical surface). E.g. add a graphical representation.

Response: Thank you for this important observation. In the simulation, cohesive elements were used between plies to model behavior of interfaces or bonds between plies under normal and tangential stresses. A statement has been added to the beginning of section 5 in the latest manuscript describing this.

More details need to be given of the fabrication of the material configuration, so how can other reproduce it?

Response: Thank you for your insightful comment. We would like to clarify that the scope of our study is centered on numerical simulations of ballistic impacts using finite element analysis (FEA). The objective was to evaluate the ballistic performance of polymer matrix composite panels with polyurea coatings under defined conditions, rather than to physically fabricate and test the materials.

To ensure reproducibility of our numerical results

  • We have explicitly detailed all material properties, boundary conditions, and model setup parameters in the latest manuscript.
  • Key assumptions and simplifications adopted in the simulation have been stated clearly.
  • The Johnson-Cook parameters, cohesive element properties, and other critical model definitions have been provided to facilitate accurate replication of the numerical study.

Furthermore, the material properties used in the model (e.g., stiffness, fracture energy, and interface properties) were derived from existing experimental data in the literature. The CFRP laminae properties were obtained from Simulia. However, the properties of GFRP, were computed using the rule of mixtures. Appropriate references to these sources have been included in section 2 of the latest submission.

Please explain in more detail the simulation approach? Add a flow chart.

Response: Thank you for your constructive suggestion. We understand the importance of a clear and transparent explanation of our simulation approach to ensure reproducibility and clarity for readers. In our study, the simulation was conducted using Abaqus finite element software to evaluate the ballistic performance of polymer matrix composite panels with polyurea coatings.

To further improve clarity, we have included a flow chart summarizing our simulation workflow in section 2 of the revised manuscript. This diagram visually represents the simulation stages, highlighting critical processes and dependencies.

Please make sure all abbreviations are explained.

Response: Thank you for identifying this oversight. We have ensured that all abbreviations are defined in the latest manuscript.

Equation (6) and (7): how reliable are these?

Equations (6) and (7) are derived from the well-established Johnson-Cook model, which has been extensively validated in both numerical and experimental ballistic studies. These equations are standard in modeling the dynamic response of metallic materials under high strain-rate conditions. Their reliability is supported by their widespread use in the literature and their validation in our independent verification step (Section 3). We have also added a reference to support this view in section 2.

Figure 3: 400 m/s; could this be a variable?

Response: Thank you for this thoughtful question. In the current study, the projectile velocity was set at 400 m/s to represent a standard ballistic impact scenario, and this value aligns with prior experimental benchmarks (e.g., Kasano [28]) which was the basis for validating our model. As we have now noted in the revised conclusion, the velocity could indeed be a variable in future investigations as design specifications are created. The results at 400 m/s provide meaningful initial insights into the ballistic response of the materials.

Reviewer 3 Report (New Reviewer)

Comments and Suggestions for Authors

The authors need to address the following points.

·         Abstract needs to be improved significantly.

·         The introduction lacks the necessary background information on the study or problem. The introduction looks like an experimental notebook.  

·         The literature review should be enhanced with quantitative data. Additionally, include a paragraph at the end of the introduction to explain how the study addresses a knowledge gap.

·         “these materials are displayed in Table I.”  Correct it as Table 1. Check the similar changes in entire manuscript.

·         For citation use author names too. Not only by [29] or in [21].

·         Maintain uniformity in citing figures and tables.

·         Tables and figures need more discussion.

·         Results and discussion need to be supported by reasons. In the results, the author(s) is expected to try as much as possible to compare their results obtained with the existing or similar studies.

·         Kindly reconcile the conclusion with the study objectives. The conclusion looks like the discussion part.

·         What are the practical implications of this study and the future directions? Kindly state?

·         The article must undergo language proofreading, as I found grammatical and typographical errors.

Comments on the Quality of English Language

The article must undergo language proofreading, as I found grammatical and typographical errors.

Author Response

The authors need to address the following points.

Abstract needs to be improved significantly.

Response: The abstract has now been significantly improved in the latest submission. The abstract now reads:

Abstract: The protective effect of polyurea (PU) coatings on Polymer Matrix Composite (PMC) panels subjected to high-velocity ballistic impacts, particularly as a potential replacement ma-terial for Large Power Transformer (LPT) tanks, has not been extensively reported in the literature. This study addresses the gap by presenting a numerical investigation into the ballistic performance of PMC panels with PU coatings. Due to the complex nature and high cost of experimental testing, this research relies on finite element modeling to predict the panels' responses under impact. Glass fiber/epoxy and carbon fiber/epoxy composite panels were tested individually and in hybrid configurations while being subjected to simulated 400 m/s steel projectile impacts. The study first investigates the impact damage evolution in uncoated panels, analyzing the arrest depth as a function of panel thickness. It then evaluates the effect of PU coatings on the ballistic response. The results demonstrate that PU coatings are three times more effective in protecting both glass and carbon fiber panels from penetration compared to simply increasing panel thickness. Additionally, the utilization of PU coatings led to a reduction in cost, mass, and thickness, while still preventing penetration of the projectile in the models.

The introduction lacks the necessary background information on the study or problem. The introduction looks like an experimental notebook.

Response: Thank you for your feedback. We appreciate your insight, and we understand the importance of providing a strong background to properly frame the study. We acknowledge that the current introduction might have appeared too focused on the experimental context, and we have now restructured it to better emphasize the numerical simulation focus and provide a clearer context for the research. Most simulation details have been moved to section 2 of the latest manuscript.

The literature review should be enhanced with quantitative data. Additionally, include a paragraph at the end of the introduction to explain how the study addresses a knowledge gap.

Response: Thanks, the paragraph below has been adopted at the end of the introduction of the latest manuscript as seen below:

In addition, our work addresses a critical gap in understanding the use of PMCs with PU coatings for ballistic impact protection, particularly in applications such as LPT tanks. Although previous studies have explored the ballistic and blast resistance of PMCs and PU coatings sepa-rately, their combined performance under high-velocity impacts has been insufficiently studied. The current research leverages a validated FE approach to provide a flexible and efficient method that reduces reliance on costly and time-consuming experimental testing. This novel numerical framework enables the exploration of the synergistic effects of PU coatings on PMC panels, of-fering insights into potential cost-effective and lightweight protective structures. The findings from this study not only contribute to the optimization of PMC panels for various structural ap-plications but also provide a new pathway for designing composite materials for ballistic protection in fields where performance and cost are critical factors.

Furthermore, additional quantitative data has been added to the literature review in the introduction.

these materials are displayed in Table I.”  Correct it as Table 1. Check the similar changes in entire manuscript.

Response: Thanks for carefully identifying our inconsistencies. They have been corrected in the latest manuscript.

For citation use author names too. Not only by [29] or in [21].

Response: Thank you for the recommendation; this has been done as well.

Maintain uniformity in citing figures and tables.

Response: Noted with thanks, these have been made consistent in the latest manuscript.

Tables and figures need more discussion.

Response: Thank you for highlighting this. We added a few comments to further explain what figures 1 and 2 represent. Also, tables 1, 2, and 3 have been discussed in detail with more content. The tables highlighting the results such as tables 4, 5, and 6 were discussed in the result and conclusion with more content.

Results and discussion need to be supported by reasons. In the results, the author(s) is expected to try as much as possible to compare their results obtained with the existing or similar studies.

Response: Thank you. We provided potential reasons for discrepancies between our findings and previous studies, considering differences in material properties, impact conditions, and simulation assumptions. We were unable to identify any published studies on polyurea coated thick CFRP laminates in the literature for comparison.

Kindly reconcile the conclusion with the study objectives. The conclusion looks like the discussion part.

Response: Thank you for bringing this to our notice. The conclusion has been rewritten to address this concern. Here is the revised conclusion:

This study aimed to numerically assess the ballistic performance of Polymer Matrix Composite (PMC) panels with polyurea (PU) coatings, particularly for their potential application in Large Power Transformer (LPT) tanks. The primary objectives were to: (1) evaluate the impact resistance of CFRP and GFRP laminates, (2) investigate the effectiveness of hybrid composite configurations, and (3) explore the benefits of PU coatings in enhancing ballistic protection while reducing cost, mass, and thickness. The results show that PU coatings significantly improve the ballistic resistance of PMC panels, proving to be three times more efficient than increasing panel thickness alone. Hybrid laminates combining CFRP and GFRP with PU coatings demonstrated improved performance, offering a balance of strength and cost-effectiveness. Moreover, the addition of PU coatings reduced both the mass and cost of the panels, while providing comparable or superior protection compared with uncoated composites.

These findings offer valuable insights into the potential use of PMC panels with PU coatings for LPT tank protection, presenting a promising alternative to traditional, heavier materials. The numerical framework developed in this study provides a reliable tool for initial material selection and design optimization in the development of protective structures for the power grid. While future work can explore additional projectile velocities and include experimental validation to confirm the recommended design, the results presented here demonstrate that PU coatings enhance ballistic resistance while simultaneously reducing mass, thickness, and cost. This marks a significant advancement in material selection and design for protective applications.

What are the practical implications of this study and the future directions? Kindly state?

Response: Thank you for your valuable feedback. We have included this in the revised conclusion. Here are some future directions we are considering but presented only two in the manuscript for the sake of brevity.

Future Directions:

While this study has provided valuable insights into the ballistic performance of PMC panels with PU coatings, future research will need to focus on various areas:

  • Experimental Validation: Future work would focus on validating the numerical results obtained in this study through physical testing of CFRP, GFRP, and hybrid composites with PU coatings, under a variety of impact conditions and temperatures. This would help confirm the effectiveness of PU coatings in real-world applications.
  • Performance under Variable Impact Conditions: The current study modeled the impact of a single projectile velocity (400 m/s). Future studies could explore the effect of varying impact velocities, projectile shapes, and multi-impact scenarios to assess the robustness and durability of PU-coated PMCs in more dynamic environments.
  • Advanced Coatings: Further investigation into the development of advanced coatings that could improve the energy absorption and fracture resistance of PMCs is another promising direction. Exploring multi-layer coatings or coatings with reinforced properties could provide enhanced protection against high-velocity impacts.
  • Modeling Under Different Environmental Conditions: Expanding the simulation framework to include different environmental conditions, such as temperature variations, moisture effects, and aging of the materials, would help assess the long-term effectiveness of PU-coated PMCs in diverse real-world conditions.
  • Application to Other Critical Infrastructure: Beyond LPT tanks, the findings could be extended to other areas requiring lightweight ballistic protection—such as military applications, aircraft protection, and civil infrastructure—where the use of composite materials combined with protective coatings could offer significant advantages.

The article must undergo language proofreading, as I found grammatical and typographical errors.

Response: Thank you, we have grammar-checked the current version of the manuscript.

Reviewer 4 Report (New Reviewer)

Comments and Suggestions for Authors

Dear Authors:

I feel your manuscript is very interesting, and if you could write it in a compelling way, it would demonstrate a high-quality paper that could attract many readers. From this perspective, please consider the following revisions:
#1: Introduction The introduction is well-written; however, I recommend citing more relevant studies in the field to strengthen the theoretical background and contextual framework.

#2: Methods While the methodology is generally described well, the paper would benefit from more detailed information on simulation parameters. Specifically, please provide additional context on:

  • Mesh size
  • Boundary conditions
  • Justification for material model choices (e.g., rationale behind using Hashin and Puck criteria)

These additions would significantly improve the reproducibility and scientific rigor of the research.

#3: Discussion and Conclusion

  • Ensure consistent presentation of units, symbols, and terminology throughout the paper.
  • Carefully proofread to eliminate grammatical and typographical errors, enhancing the manuscript's clarity and professional appearance.

Overall, this is a strong and impactful paper that addresses a critical gap in materials science and engineering. With the suggested refinements, the manuscript will become even more compelling and accessible to readers in the field.

Comments on the Quality of English Language

I recommend a thorough proofreading of the manuscript, preferably by a native English speaker or a professional editor, to further refine the language. This careful review will help ensure that the paper meets the high standards expected for publication.

Author Response

I feel your manuscript is very interesting, and if you could write it in a compelling way, it would demonstrate a high-quality paper that could attract many readers. From this perspective, please consider the following revisions:

Response: Thank you very much.

#1: Introduction The introduction is well-written; however, I recommend citing more relevant studies in the field to strengthen the theoretical background and contextual framework.

Response: Thank you very much for your valuable feedback. We added more references to the introduction. We also moved experimental details out of the introduction to section 2.

#2: Methods While the methodology is generally described well, the paper would benefit from more detailed information on simulation parameters. Specifically, please provide additional context on:

Mesh size.

Boundary conditions

Response: Thank you once again for your encouraging comments. To provide further clarity on the context of the mesh size and boundary conditions we have included Figure 3 in the revised manuscript. Figure 3 visually illustrates the types of meshes applied to different regions of the simulation geometry and the boundary conditions utilized. Details of the specific mesh sizes are discussed at appropriate points in the revised manuscript.

Justification for material model choices (e.g., rationale behind using Hashin and Puck criteria)

Response: Thank you for your valuable observation. The justification for using Hashin, and Puck stems from the fact that unidirectional (UD) PMC were used in our simulation. UD PMCs were used because the composite properties are strongly influenced by ply orientation, stacking sequence, and ply thickness. To prevent undesired bending and twisting in the laminates when used as a structural material (e.g., in Large Power Transformer (LPT) tanks), specific conditions must be met in the material’s stiffness matrices.

  • In particular, the A-matrix elements A16 = A26 must be set to zero. This ensures that normal strains caused by normal stresses are uncoupled from shear strains caused by shear stresses, preventing twisting of the laminate during operational loading.
  • Furthermore, to prevent bending, the B-matrix must also equal zero, ensuring that the material does not bend as it strains during loading.
  • Additionally, to avoid coupling between bending and twisting deformations, it is critical that the D-matrix elements D16 = D26 = 0

In composite design, these criteria are typically achieved by creating balanced and symmetric laminates. However, to eliminate any uncertainty regarding potential complications, unidirectional CFRP was chosen, as it ensures no coupling effects across all loading conditions.

Furthermore, a statement summarizing these concepts was added to section 2 of the latest manuscript.

These additions would significantly improve the reproducibility and scientific rigor of the research.

Response: Thanks, and that is very correct.

#3: Discussion and Conclusion

Ensure consistent presentation of units, symbols, and terminology throughout the paper.

Response: Thank you for your comment. This has been implemented in the revised manuscript.

Carefully proofread to eliminate grammatical and typographical errors, enhancing the manuscript's clarity and professional appearance.

Thanks again, we have grammar-checked the latest manuscript.

Overall, this is a strong and impactful paper that addresses a critical gap in materials science and engineering. With the suggested refinements, the manuscript will become even more compelling and accessible to readers in the field.

Thank you.

I recommend a thorough proofreading of the manuscript, preferably by a native English speaker or a professional editor, to further refine the language. This careful review will help ensure that the paper meets the high standards expected for publication.

Thanks, and done.

Round 2

Reviewer 2 Report (New Reviewer)

Comments and Suggestions for Authors

The authors have made changes to an acceptable extend, although I advise them for future work to increase the change level regarding comments made.

Reviewer 3 Report (New Reviewer)

Comments and Suggestions for Authors

The authors have addressed all the queries. The article may be accepted in its current form.

This manuscript is a resubmission of an earlier submission. The following is a list of the peer review reports and author responses from that submission.

Round 1

Reviewer 1 Report

Comments and Suggestions for Authors

Review report of “Impact Resistance of Potential Replacement Materials for Large Power Transformer Tanks” by Williams et al.

In this manuscript, the authors explored the impact resistance of polymer matrix composites (PMCs) as a potential replacement for low-carbon steel in LPT tanks. The study employed finite element simulations to compare the ballistic performance of LCS and PMCs, with and without a polyurea coating. This is a useful work, but it requires substantial revision to address numerous issues before publication.

1.   The manuscript fails to clearly articulate the novelty of the work. Much of the content reiterates known concepts in polyurea and its usage in impact protection. Indeed, in Page 4, Section 1.4, the authors made a note that [NEED PARAGRAPH ABOUT NOVELTY OF THIS WORK]. The authors should add novelty of their work before submission. 

2. The manuscript lacks a rigorous FEM framework. The choice of simulation parameters, and failure models seem arbitrary and insufficiently justified. For example, there are many mechanics models for polyurea in literature. The rationale behind selecting certain material model for the simulations is not adequately explained. Since the conclusion of the paper heavily relies on numerical simulations without experimental validation, the authors should discuss, compare, and properly select the material models.

3. The assumptions made in the simulations are overly simplified, which significantly undermines the validity of the conclusions. The authors fail to account for the complex, real-world conditions that LPT tanks are exposed to, such as varying environmental factors and mechanical stresses.

4. The literature review is insufficient, with several seminal references in polyurea impact resilient missing. Some recent review and research papers about impact resilience of polyurea under high strain rates should be cited:

(1) Zhang, R. et al. (2022). Polyurea for blast and impact protection: A review. Polymers, 14(13), 2670.

(2) Jin, H. et al. (2022). Dynamic fracture of a bicontinuously nanostructured copolymer: A deep-learning analysis of big-data-generating experiment. Journal of the Mechanics and Physics of Solids, 164, 104898.

(3) Kim, H. et al. (2012). Dynamic fracture energy of polyurea-bonded steel/E-glass composite joints. Mechanics of Materials, 45, 10-19.

5. VUMAT is just an Abaqus subroutine. It is well defined as “User subroutine” in ABAQUS. You can define any models in VUMAT. It should not be called "virtual user material". 

6. Proper citations for Fig. 1(a) should be given unless it’s made by the authors. What is the purpose for showing the geometry in Fig.1(b)? 

Author Response

Reviewer's comment 1: The manuscript fails to clearly articulate the novelty of the work. Much of the content reiterates known concepts in polyurea and its usage in impact protection. Indeed, in Page 4, Section 1.4, the authors made a note that [NEED PARAGRAPH ABOUT NOVELTY OF THIS WORK]. The authors should add novelty of their work before submission.

Author’s Response: Thank you for your comment. Based on your feedback, we have significantly improved the paper by focusing on the application of polyurea PU as a potential coating for ballistic protection of polymer matrix composites (PMCs) that could be used for large power transformer tanks (LPTs). We believe this study for this application has never been done before. In addition, we studied the effect of different composite architectures for two PMCs: GFRP and CFRP in combination with PU. For emphasis, we are not focusing on blast resistance per se; we are looking at the penetration resistance from riffle bullets on the combination of PU and PMCs in a new application in LPT tanks. The capabilities of PMCs as a potential replacement for low-carbon steel have been rigorously verified in a case study published by the same authors [1].

Reviewer's comment 2: The manuscript lacks a rigorous FEM framework. The choice of simulation parameters and failure models seems arbitrary and insufficiently justified. For example, there are many mechanics models for polyurea in literature. The rationale behind selecting certain material model for the simulations is not adequately explained. Since the conclusion of the paper heavily relies on numerical simulations without experimental validation, the authors should discuss, compare, and properly select the material models.

Author’s response: Thank you for your valuable feedback. Firstly, the concerns have been addressed in section 2.2. PU properties and failure model in the revised version of the manuscript. More details have been added to accommodate all the concerns highlighted in the reviewer’s comment concerning the modeling quality.

Reviewer's comment 3: The assumptions made in the simulations are overly simplified, which significantly undermines the validity of the conclusions. The authors fail to account for the complex, real-world conditions that LPT tanks are exposed to, such as varying environmental factors and mechanical stresses.

 Author's Response: Indeed! The reviewer was right in pointing out the complex real-world exposures of these tanks however, this was extensively discussed in our first paper related to this subject published in IEEE Power Delivery [2], which was cited in the manuscript. Stating all the issues in the manuscript may detract from the main point of focus which is the impact protection of potential polymeric composites in LPTs. In addition, the simplified models used in the manuscript were intentional for the following reasons:

  • The complexity of combining many quantities and variables in ballistic modeling, with a unique simulation for varying thickness, and architecture of polymer composites makes it difficult to keep track of key variables such as velocity and penetration distances in describing the effect of PU on ballistic impact.
  • Based on the first point above we ensured that the assumptions did not violate the physics of the problem using well-proven and published models.

We acknowledge that more detail could add further nuance to the already presented results, but this might obscure the primary takeaways of this research. The aim of this work in our opinion is to create a balance between theoretical rigor and concept clarity so that the reader can understand the novelty of using PU coating on a potential material replacement for LPT tanks (which is novel) without being overwhelmed by the complexities involved in the modeling which may detract them from the primary focus on the paper.

Please note that the idea behind using polymer matrix composites as the structural material in LPT tanks has been rigorously discussed with the US Department of Energy and has been published in IEEE by our team [2].

Reviewer's comment 4: The literature review is insufficient, with several seminal references in polyurea impact resilient missing. Some recent review and research papers about impact resilience of polyurea under high strain rates should be cited:

Author’s response: Thank you for making us aware of these additional sources. All the articles suggested have been appropriately included in the latest version of the manuscript. In addition, this work references our first paper which extensively described the intricacies of using polymeric composites for the LPT tank.

Reviewer's  Comment 5: VUMAT is just an Abaqus subroutine. It is well defined as “User subroutine” in ABAQUS. You can define any models in VUMAT. It should not be called "virtual user material".

Author’s response: Thank you for pointing this out. We have corrected it in the revised manuscript attached.

Reviewer 2 Report

Comments and Suggestions for Authors

This manuscript is in draft stage (poor affiliations, "[NEED PARAGRAPH ABOUT NOVELTY OF THIS WORK]"-introduction, "[REF]"line 237, figure 3 no reference data - simulations difference in legends/colours very poor!, [Could include: It is interesting to note that even if the critical strain on PU were set 351 at the bottom of the 0.2-0.3 range, the required thickness of PU to prevent penetration in 352 the simulation only increased to 4 mm. This demonstrates that PU could be effective even 353 with conservative parameters.] - line 351-354), has lots of draft related comments in the text of the manuscript. Because of that I recommend a rejection and resubmission in different journal since the quality is not achieved for Polymers in the work. 

Please consider these;

-Less subheadings, more structured manuscript.

-Line 511 pg 18; If it has already been presented, why is this paper important? Please conclude with the importance of this manuscript rather than previous works and it is not great to use references-previous work in the abstract. Please either explain it better, or disconnect from the previous work.

Comments on the Quality of English Language

Minor English editing is needed but more importantly a proof-reading for final version is needed since the current version is in draft stage. 

Author Response

Reviewer's comment:  This manuscript is in draft stage (poor affiliations, "[NEED PARAGRAPH ABOUT NOVELTY OF THIS WORK]"-introduction, "[REF]"line 237, figure 3 no reference data - simulations difference in legends/colours very poor!, [Could include: It is interesting to note that even if the critical strain on PU were set 351 at the bottom of the 0.2-0.3 range, the required thickness of PU to prevent penetration in 352 the simulation only increased to 4 mm. This demonstrates that PU could be effective even 353 with conservative parameters.] - line 351-354), has lots of draft related comments in the text of the manuscript. Because of that I recommend a rejection and resubmission in different journal since the quality is not achieved for Polymers in the work.

Please consider these;

-Less subheadings, more structured manuscript.

-Line 511 pg 18; If it has already been presented, why is this paper important? Please conclude with the importance of this manuscript rather than previous works and it is not great to use references-previous work in the abstract. Please either explain it better, or disconnect from the previous work.

Author’s response:  Thank you for your candid feedback. We have made a significant revision to the manuscript, focusing on PMCs only (instead of PMCs and low-carbon steel). This simplifies the paper and allows more relevant details to be included.

Round 2

Reviewer 1 Report

Comments and Suggestions for Authors

Th authors have addressed my comments. The paper can be accepted now.

Reviewer 2 Report

Comments and Suggestions for Authors

As authors declared, significant revision has been made in the manuscript. Therefore, a new review process must be started after the rejection. The subheadings are still in there. Authors should have proof-checked and applied the comments provided. 

Comments on the Quality of English Language  

Minor English editing is needed but more importantly a proof-reading for final version is needed.